# Intestinal Epithelium-Derived Luminally Released Extracellular Vesicles in Sepsis Exhibit the Ability to Suppress TNF-α and IL-17A Expression in Mucosal Inflammation

**DOI:** 10.3390/ijms21228445

**Published:** 2020-11-10

**Authors:** Michael G. Appiah, Eun Jeong Park, Samuel Darkwah, Eiji Kawamoto, Yuichi Akama, Arong Gaowa, Manisha Kalsan, Shandar Ahmad, Motomu Shimaoka

**Affiliations:** 1Department of Molecular Pathobiology and Cell Adhesion Biology, Mie University Graduate School of Medicine, Tsu, Mie 514-8507, Japan; 317DS06@m.mie-u.ac.jp (M.G.A.); 316DS12@m.mie-u.ac.jp (S.D.); a_2.uk@mac.com (E.K.); y-akama@clin.medic.mie-u.ac.jp (Y.A.); arong-g@doc.medic.mie-u.ac.jp (A.G.); 2Department of Emergency and Critical Care Center, Mie University Hospital, Tsu, Mie 514-8507, Japan; 3School of Computational and Integrative Sciences, Jawaharlal Nehru University, New Delhi 110067, India; manisha8013@gmail.com (M.K.); shandar@jnu.ac.in (S.A.)

**Keywords:** sepsis, intestinal epithelial cells, inflammation, extracellular vesicles, TNF-α, IL-17A, miRNAs

## Abstract

Sepsis is a systemic inflammatory disorder induced by a dysregulated immune response to infection resulting in dysfunction of multiple critical organs, including the intestines. Previous studies have reported contrasting results regarding the abilities of exosomes circulating in the blood of sepsis mice and patients to either promote or suppress inflammation. Little is known about how the gut epithelial cell-derived exosomes released in the intestinal luminal space during sepsis affect mucosal inflammation. To study this question, we isolated extracellular vesicles (EVs) from intestinal lavage of septic mice. The EVs expressed typical exosomal (CD63 and CD9) and epithelial (EpCAM) markers, which were further increased by sepsis. Moreover, septic-EV injection into inflamed gut induced a significant reduction in the messaging of pro-inflammatory cytokines TNF-α and IL-17A. MicroRNA (miRNA) profiling and reverse transcription and quantitative polymerase chain reaction (RT-qPCR) revealed a sepsis-induced exosomal increase in multiple miRNAs, which putatively target *TNF-α* and *IL-17A*. These results imply that intestinal epithelial cell (IEC)-derived luminal EVs carry miRNAs that mitigate pro-inflammatory responses. Taken together, our study proposes a novel mechanism by which IEC EVs released during sepsis transfer regulatory miRNAs to cells, possibly contributing to the amelioration of gut inflammation.

## 1. Introduction

Sepsis is a life-threatening organ dysfunction resulting from a dysregulated host immune response to infection [1]. Sepsis is a principal cause of mortality in the intensive care unit (ICU) setting, with an estimated 48.9 million case incidence and 11 million deaths occurring globally in 2017 [2]. The major causes of mortality and morbidity stem from multiple organ failure. The gut is one of the major organs damaged during sepsis. Gut epithelial-barrier dysfunction is a primary pathological feature in sepsis and has been shown to be detrimental to survival [3]. In sepsis, splanchnic hypoperfusion causes gut injury, which in turn results in the release of gut-derived pro-inflammatory factors [4,5] that reach the systemic circulation through the mesenteric lymph [6,7,8,9]; a process thought to exacerbate and maintain an inflammatory response that culminates in multiple organ failure [10]. Thus, the gut has been described as the “motor” of multiple organ dysfunction syndrome (MODS) both in sepsis and other critical illnesses, due to its potential role as a portal for generating systemic immune responses [11]. Infectious and inflammatory injury to the intestinal epithelial cells not only compromises barrier functionality, but also induces the secretion of pro-inflammatory and regulatory cytokines [12,13,14]. Pro-inflammatory cytokines secreted into the intestinal luminal space during sepsis are thought to act as important regional mediators that complicate the pathology of intestinal failure [14]. As detailed in the following sections, we have explored the role of extracellular vesicles (EVs) as an alternative/additive regional mediator to luminally secreted cytokines in sepsis.

Small EVs called exosomes are biologically functional nanoparticles released from virtually all cells including intestinal epithelial cells [15]. EVs serve as mediators of intercellular communication between both neighboring and distant cells. EVs carry and directionally transfer a repertoire of bioactive molecules, such as proteins, lipids, and genetic materials (e.g., mRNA, miRNA), all of which are capable of modifying the functions of the recipient cells [16]. EVs secreted from immune cells, platelets, and endothelial cells have been shown to play important roles in the pathogenesis of sepsis. Dendritic cells have been shown to produce EVs that abrogate septic inflammatory responses by delivering milk fat globule EGF factor VIII (MFG-E8) [17]. EVs derived from circulating platelets have been reported to aggravate multiple organ dysfunction during sepsis by transferring reactive oxygen species and pro-inflammatory miRNAs to their target cells [18]. Endothelial exosomes in septic mice have been shown to exert their protective effects against cardiomyopathy by delivering heat shock protein [19]. In addition, exosomes in the plasma of septic patients, which are likely to be derived from multiple cells such as leukocytes, platelets, and vascular endothelial cells, have been shown not only to contain miRNAs related to inflammation and cell-cycle regulation [20], but also to express the programmed cell death 1 (PD-1) ligand [21].

IECs have been shown to secrete EVs in two alternative directions, basolaterally and apically. Chen et al. have shown that basolaterally released IEC EVs carry food antigens while expressing αVβ6 integrin that activates TGF-β thereby inducing the tolerogenic capacities of dendritic cells in the lamina propria space [22]. Kojima et al. have shown in a rat model of trauma and hemorrhagic shock that basolaterally released IEC EVs, with upregulated apoptosis-inducing Fas ligand expression, were drained to the mesenteric lymph nodes, thereby inhibiting the functions of dendritic cells and lymphocytes [23]. By contrast, using an IEC line infected by a protozoan parasite, Hu et al. have shown that exosomes released from IECs to the apical side carry antimicrobial peptides, thereby playing a role in host defense [24]. Deng et al. have shown in vivo that apically released IEC EVs isolated from the intraluminal space of the mouse intestine carry prostaglandin E2 to suppress natural killer T-cell functions [25]. In addition, using colonic fluid aspirates from inflammatory bowel disease patients, Mitsuhashi et al. have shown that apically released IEC in the intraluminal space become enriched with pro-inflammatory cytokines [26].

These previous reports point to the potential regulatory roles of IEC-released EVs; however, their nature and immuno-modulatory effects in sepsis remain to be elucidated. Here, we have studied the IEC-derived EVs released into the intestinal luminal space, demonstrating their actions to suppress TNF-α and IL-17A expression in intestinal mucosal inflammation in vivo. Luminally-released IEC EVs in sepsis are thought to act in a para- and autocrine manner to counter-balance inflammation and, thus, contribute to the maintenance of regional immune-homeostasis.

## 2. Results

### 2.1. Luminal Lavage EVs Are Epithelial Derived and Express Exosome Markers

Using differential ultracentrifugation, EVs were isolated from the intestinal lavage fluids of septic mice 24 h after CLP induction in mice. As a control, EVs were isolated from the intestinal lavage fluids of sham-operated (laparotomy) mice 24 h after the procedure. By this time septic mice manifested shock symptoms such as lethargy, piloerection, periorbital exudate, and diarrhea [27] and by 72 h were all dead (Appendix A). Compared to those of control mice, mRNA levels of the IL-6, IL-10, and CCL3 in peripheral blood mononuclear cells (PBMCs) of septic mice were significantly increased (Appendix A), all of which are known to be key markers of sepsis severity and mortality [28,29].

EVs isolated from large intestinal lavage fluid expressed the epithelial marker EpCAM, but not the leukocyte marker CD45 or the platelet maker CD41 (Figure 1A). Thus, the majority of EVs were derived from gut epithelial cells, while only a small proportion was presumably from the circulation. EVs isolated from the large intestine lavage fluid showed the typical exosomal markers CD63 and CD9 (Figure 1A). In contrast, EVs from the small intestine lavage fluid showed only CD63 but lacked CD9 and EpCAM-1 expression (Appendix A). In some experiments, EVs isolated via the more stringent C-DGUC method showed similar results (Appendix A). Therefore, EVs from the large intestine were classified as belonging to the typical exosome category while those from the small intestine were excluded. For the sake of simplicity and clarity, in the sections that follow we focus on the EVs from the large intestine lavage fluid, although we are aware that some of these EVs isolated may have been secreted from the small intestine.

Expression of the tetraspanins CD63 and CD9 from large intestine lavage EVs was upregulated during sepsis (Figure 1A). Moreover, EpCAM-1 on those EVs was also enhanced by sepsis (Figure 1A). Thus, sepsis-induced upregulation of typical exosomal and epithelial marker expressions may reflect the possibility that luminal release of IEC-derived EVs is facilitated by the pathological condition of sepsis.

The sizes and particle densities of the EVs were measured using nanoparticle tracking analysis (NTA) or dynamic light scattering (DLS). The EVs were found to range from 50 to 600 nm and displayed a major peak at approximately 200 nm (Figure 1B). The density of EVs from sepsis mice was 1.10 ± 0.16 × 10^9^ while that from sham-treated mice was 0.82 ± 0.17 × 10^9^ particles per mL (Figure 1C). The average sizes of the EVs were 157 ± 17 nm and 208 ± 26 nm for sepsis and control mice, respectively (Figure 1D).

As gut lumen harbors numerous microflora [30], the EVs might contain bacterial EVs. Thus, we sought to measure the level of bacteria endotoxin (lipopolysaccharide; LPS) that is contained in the EVs secreted from gram-negative bacteria. Measurements by ELISA revealed that LPS levels in all EV samples from the luminal lavage of mice and from the fecal samples of septic patients were similar to those of a negative control (BSA), and much lower than those in the plasma and feces of septic mice (Appendix A). Notably, the plasma samples of septic mice showed increased LPS levels. Therefore, LPS levels were barely detectable in the EV samples. Although it is still possible that trace amount of bacterial and/or fungal components were present, IEC-derived exosomes constituted the majority of the EV samples isolated from the large intestinal lavage fluids in this study.

### 2.2. Luminal Lavage EVs of Sepsis Mice Regulate Pro-Inflammatory Cytokine Expression in a Colitis Model

To test our research hypothesis that intestinal epithelial EVs function as a para- and autocrine regulator of mucosal immunopathology, we first performed ileal-loop assay in which luminally secreted intestinal epithelial EVs were injected into the luminal space of the tied ileum of healthy mice, and in a separate experiment, the tied ileum of DSS-induced colitis mice. Although DSS-induced colitis is not directly relevant to sepsis-induced mucosal immunopathology, we reasoned that mucosal inflammation in DSS-induced colitis could serve as an in vivo bioassay for studying the ability of luminally-secreted intestinal epithelial EVs to modulate regional immunological responses. Mucosal inflammation was induced by the oral administration of 2.5% DSS-containing drinking water for seven days. This in vivo bioassay system could recapitulate a scenario in which luminally secreted intestinal epithelial EVs act on mucosal inflammation. Eighteen hours after EV treatment, the mice were sacrificed and the intestinal tissues were harvested and subjected to quantitative RT-PCR analysis of some key pro-inflammatory mediators in both septic [28,31] and DSS-induced mucosal inflammation [32,33].

In the ileal tissues of healthy mice, sepsis EVs (S-EVs) downregulated messages of the pro-inflammatory cytokines including TNF-α, IL-1β, IL-6, IL-17A, and IL-22 (Appendix A). DSS-treatment increased the expressions of all the aforementioned cytokines in the gut tissues, as was evident in vehicle (PBS)-administered mice (Figure 2 and Appendix A). We noted that, administration of S-EVs into the ileal loop of DSS-colitis mice significantly suppressed the mucosal inflammation-associated increase in TNF-α and IL-17A compared to that of control EVs (C-EVs) (Figure 2). Contrasting effects were, however, seen between both EVs with regards to IL-1β, IL-6, and IL-22 as their expressions were significantly decreased by S-EVs but increased by C-EVs (Appendix A). We compared these results with those which were isolated by using the cushioned-density gradient ultracentrifugation (C-DGUC) which is another method offering improved purity to EVs [34,35], in order to validate biological effects of the EV samples isolated by using differential UC. As a result, treatment with S-EVs isolated by C-DGUC was also shown to reverse the upregulation of TNF-α and IL-17A in the DSS-induced gut inflammation (Appendix A), suggesting that S-EVs may possess a healing effect in the inflamed intestinal tissues.

We next sought to confirm, using in vitro system, the regulatory effects of S-EVs on gut expression of pro-inflammatory mediators. Based on our hypothesis that gut epithelium is the main target affected by S-EVs, we isolated primary gut ECs, confirmed their expression of EpCAM (Appendix A), and then treated with different EV doses (1, 5, or 10 µg/mL) to determine the impacts on regulating TNF-α and IL-17A expression. But, S-EVs as well as C-EVs didn’t exhibit any significant reduction, at any EV concentration used, in TNF-α and IL-17A expression of primary gut ECs, when compared with vehicle treatment (Appendix A). Accordingly, establishment of more physiologically relevant models might be beneficial to achieve in vitro outcome that correlates with in vivo result. We have discussed more about this matter in Section 3 below.

### 2.3. Luminal Lavage EVs in Sepsis Enrich the miRNAs Regulating Inflammation

Here, we investigated the ability of intestinal epithelial EVs in sepsis to suppress TNF-α, and IL-17A based on the assumption that such a capability would primarily be mediated by exosomal miRNAs. To this end, we conducted in silico analysis of exosomal miRNAs. To compare miRNA profiles of the intestinal epithelial EVs in septic mice versus those of control mice, we performed small-RNA deep sequencing, identifying a total of 417 miRNAs based on criteria stipulating a value of ≥1 reads per kilobase of transcript per million mapped reads (RPKM) (Appendix A). Of the 417 miRNAs, 45 and 190 miRNAs were expressed exclusively in control (C-EVs) and sepsis EVs (S-EVs), respectively, while 182 miRNAs were expressed in both (Figure 3A). Moreover, 292 miRNAs were upregulated, whereas 68 miRNAs were downregulated in S-EVs with a fold change of ≥2 compared to control EVs (Figure 3B). Gene ontology (GO) enrichment analysis of targets of the sepsis-increased miRNAs revealed transcription regulation-related GO molecular functions. This was corroborated by the enriched transcription-related GO biological process terms (Appendix A and data not shown). In addition, GO cellular compartments were enriched with nucleus-related terms (Appendix A).

Using a TargetScan bioinfomatics analysis that concentrated on miRNAs upregulated in the intestinal epithelial EVs during sepsis, we examined those miRNAs predicted to target TNF-α, and IL-17A. We found 32 and 36 miRNAs that targeted TNF-α and IL-17A, respectively, as well as 17 others which target both genes (Figure 3C and Appendix A). We selected some of the TNF-α and IL-17A targeting miRNAs based on knowledge of their involvement in modulating inflammatory response (Figure 4A) and using RT-qPCR, we confirmed upregulation of these miRNAs in sepsis intestinal epithelial EVs, compared with control intestinal epithelial EVs (Figure 4B).

## 3. Discussion

Accumulating evidence suggests that extracellular vesicles (EVs) are involved in the pathogenesis of sepsis, with different effects based on the multiple sources and targets of EVs [36,37]. Much less is known, however, about how EVs are shed from the gut and their precise roles in this compartment, which perpetuates septic inflammatory responses. Thus, we investigated intestinal epithelial cell (IEC)-derived EVs in sepsis and their roles in inflammation. IECs secrete various cytokines and chemokines, especially under inflammatory conditions [38]. The recognition of pathogen-associated molecular patterns (PAMPs) by pattern recognition receptors (PRRs) on IECs and other cells of the mucosal immune system during sepsis stimulates the release of a plethora of pro-inflammatory cytokines, which in turn induces the infiltration of leukocytes to the gut mucosa, further exacerbating the inflammatory response [39]. This dysregulated inflammatory response may lead to vasodilatation, increased capillary leakage, and decreased oxygen supply to the intestinal epithelium, which can cause a breakdown of the epithelial barrier and foster translocation of gut bacteria and/or their products into previously sterile sites [40]. In the current study, we found that EVs from the intestinal lumen of septic mice exhibited significant changes compared to their healthy counterparts. In addition, there appeared to be a slight increase in the number of EVs during sepsis (Figure 1C). These results are consistent, at least in part, with those of a previous report that showed increased numbers of EVs in the plasma of septic mice [18].

Although the roles played by exosomes in the pathogenesis of sepsis have been studied for years, controversy persists regarding whether exosomes exert harmful or protective roles. Some previous studies found that exosomes isolated from septic mice and patients have harmful effects [41,42,43,44]. For example, exosomes derived from the platelets of septic patients exhibited a negative ionotropic effect on an isolated rabbit heart, which may be evidence of a possible link to sepsis-induced myocardial dysfunction [41]. Exosomes from septic mice suppressed the contraction of the left ventricle of the heart in healthy recipient mice [43]. Exosomes from mice suffering a septic acute lung injury induced lung inflammation in healthy recipient mice, possibly through exosomal miR-155 [44]. In addition, treatment with small molecule GW4869 that has been posited to inhibit exosome biogenesis and release, dampened sepsis-induced cardiac inflammation and dysfunction, leading to the improved survival rates in CLP-induced septic mice [42]. In contrast, other studies demonstrated that exosomes isolated from septic mice and patients exhibited protective effects [45,46]. For instance, exosomes isolated from the plasma of septic patients improved survival rates in a feces-induced sepsis mouse model, possibly via miR-7-5p which inhibits T-lymphocyte apoptosis [46]. Exosomes from septic mice alleviated the progression of lung and liver injury, thereby improving the survival rates of CLP-induced septic mice [45]. The results presented in this study have shown that luminally secreted intestinal epithelial exosomes suppresses TNF-α and IL-17A expression in inflamed mucosal tissues. Thus, our study supports the hypothesis that sepsis induces intestinal epithelial cells to secrete exosomes in order to dampen local mucosal inflammation.

The suppression of tissue TNF-α and IL-17A expression by septic luminal epithelial exosomes is thought to be mediated by exosomal miRNAs targeting the 3′-UTR of TNF-α and IL-17A mRNAs. Indeed, IECs are known to produce both TNF-α and IL-17A, which are critical players in the pathogenesis of sepsis [47,48] although other sources of TNF-α (macrophages, fibroblasts) and IL-17A (γδ T cells, ILC3) exist especially in the inflamed gut mucosa. In particular, Paneth cell-derived IL-17A has been shown to help induce systemic inflammatory responses in a model of TNF-induced shock [48]. IEC-derived exosomes released into mesenteric lymph following trauma and hemorrhagic shock were reported to possess an ability to induce acute lung injury [49]. Recently, IEC-derived exosomes were shown to induce the inflammatory responses of splenocytes in a parasite-infection model [50]. Intriguingly, in the same model but examining parasite infection with *Cryptosporidium parvum*, apically released IEC exosomes exhibited microbicidal activity against *C. parvum* [24].

The constitution of gut microbiome is altered by sepsis, which may contribute to organ dysfunction [51,52]. S-EVs are thought to be capable of eliciting inflammation by delivering cytokines or damage-associated molecular pattern to target tissues [53]. In accordance with these notions one can consider that the EVs secreted during sepsis may reshape contents of gut microbiota by direct or indirect transfer of their biologically active contents including pro-inflammatory mediators or miRNAs. In current study, IEC-derived EVs of septic mice have been proposed to play a regulatory, but not degenerating, role. Consequently, it is worthwhile to test if the EVs derived from different sources (e.g., IECs, macrophages, lymphocytes, or blood) during sepsis influence the composition of the gut microbiome [54]. Indeed, elaborate acquisition of information on relationship between S-EV-induced microbiome alteration and pathophysiologic outcome will be instrumental to design EV-based therapeutics to gut inflammation in the future.

To elicit miRNA-mediated gene silencing, exosomes administered into the intraluminal space are likely to be uptaken by those cells in the gut mucosa, recapitulating the para- and autocrine mechanisms that maintain local immune homeostasis in gut mucosal tissues. The luminal spaces of the large and small intestines contain EVs secreted not only from gut epithelial cells, but also from commensal bacteria and other microorganisms. Bacterial EVs from mice have been shown to elicit systemic inflammatory responses through Toll-like receptors 2 and 4 when intraperitoneally injected [55]. Although we have excluded the major contamination of bacterial components by measuring the endotoxin levels of our EV samples, we are aware that a trace amount of LPS and/or other microbial components might be present. For this reason, we avoided administering them intravenously in vivo. Instead, we selected the ileal-loop assay to test the roles played by the EV samples in the gut microenvironments that commensal bacteria co-habit. Interestingly, it has been shown that some of the intestinal epithelial cell-derived miRNAs do enter commensal bacteria, thereby regulating bacterial gene expression and affecting bacterial growth [56]. Thus, in addition to host cells in the gut, commensal bacteria might alternatively be subjected to EV-mediated regulation in order to maintain mucosal immune homeostasis.

We found that 292 miRNAs were upregulated in IEC-derived EVs with sepsis, an amount similar to those reported in previous studies that utilized septic clinical and mouse EVs [18,20]. The GO results from the clinical EV samples predicted miRNA profiling for inflammatory and immune responses [20]. In our GO analysis, the transcription-related molecular functions and biological processes were upregulated for enriched miRNAs (Appendix A). This disparity could have occurred, at least partly, due to differences in the experimental subjects, exosome isolation method, timeline analyses, and/or the tissues from which the exosomes were isolated.

Following the observation that S-EVs downregulate messaging of TNF-α and IL-17A in the inflamed gut, we attempted to confirm the sepsis-increased expression of several miRNAs predicted to target these genes. We realized that multiple miRNAs in luminal EVs, including miR-19a, -21, -27a, -126, 146b, and -200b were upregulated by sepsis (Figure 4). Previous reports support the possible roles of those EV miRNAs in mitigating systemic inflammatory responses. Overexpression of miR-19a reduced TNF-α in a model of LPS-induced endometritis [57]. Paclitaxel-induced amelioration of liver injury in sepsis was empowered by miR-27a-mediated downregulation of inflammatory responses [58], whereas increased miR-27a during sepsis aggravated the inflammatory response [59], demonstrating that the exact role played by this miRNA in modulating sepsis pathogenesis remains incompletely understood. In an ischemic preconditioning-polymicrobial sepsis model, exosomal miR-21 repressed NF-κB signaling and decreased pro-inflammatory cytokine production in remote organs leading to increased survival in septic mice [60]. miR-126 carried by endothelial progenitor cell exosomes protected against sepsis-induced microvascular dysfunction, lung and kidney injury, and cardiomyopathy [19,61,62]. miR-146b ameliorates LPS-induced acute lung injury [63], whereas its reduction increases IL-17A and promotes T cell acute lymphoblastic leukemia cell migration and invasion [64]. Transfection of HEK293 and THP-1 cells stably expressing TLR4 with miR-200b, and miR-200c mitigates activation of NF-κB activation and diminishes endotoxin-induced expression of TNF-α, IL-6, and other pro-inflammatory cytokines [65]. Considering all these evidence, our study supports the contention that sepsis-enhanced functional miRNAs in IEC EVs possess the ability to regulate *TNF-α* and *IL-17A* expression in the gut. However, our study lacks a functional examination of treating inhibitors for specific miRNAs in order to confirm the effects of EVs. Thus, it would be worthwhile to conduct an antagonizing and/or mimicking assay in vivo in the context of inflammations.

IL-6, IL-22, and IL-1β were significantly upregulated by C-EV compared to PBS and/or S-EV (Appendix A). Though known for their pro-inflammatory functions, IL-6 and IL-22 protect IECs from apoptosis and stimulate IEC proliferation and wound repair [66,67,68]. Thus, the definitive effect of the luminal IEC-derived EV-induced cytokine modulation on the restitution and integrity of the gut epithelial barrier under inflammatory conditions remains to be elucidated.

As aforementioned, S-EVs were not able to reduce TNF-α and IL-17A expression in the culture of isolated IECs (Appendix A). A few possible explanations for this discrepancy can be regarded. First, our in vivo data (Figure 2) indicate that S-EVs exert a regulatory effect on expression of TNF-α and IL-17A in the condition of gut inflammation. Thus, it will be worthy to verify these regulatory effects of S-EVs in gut ECs of the inflamed intestines. Second, because a majority of the isolated cells proved to be epithelial positive, it is unlikely that other immune cells such as macrophages or lymphocytes affect the outcome. Thus, one cannot rule out the possibility that any of those immune cells are sensitive to EV-mediated downregulation of expression of both mediators in immune cells. Third, it is imperative to exploit physiologically relevant condition of gut epithelia. Therefore, culturing intestinal organoids might be suitable for examination, in near future, to corroborate the regulatory effects of S-EVs, which was observed in our in vivo study.

In conclusion, we have shown that gut-lavage EVs are exclusively IEC-derived and that their expression of an epithelial marker (EpCAM) is augmented by sepsis. Sepsis EVs, compared to controls, were capable of downregulating pro-inflammatory gene expression in a gut-inflamed model. The IEC-derived EVs transformed, acquiring a highly regulatory miRNA composition following sepsis, thereby suggesting their potential role in modulating gut inflammation. This process is illustrated in Figure 5, in which septic IEC-derived luminal EVs regulate mucosal inflammatory response in the gut. Taken together, our study could provide potential clues to the previously unappreciated role played by sepsis EVs in the inflamed gut.

## 4. Materials and Methods

### 4.1. Mice

Balb/c mice (8–12 weeks old) were obtained from Japan SLC (Shizuoka, Japan). Mice were housed in the Experimental Animal Facility of Mie University. Mice were allowed to acclimatize for at least one week before being used for experiments. Mice were kept under specific-pathogen free conditions with 12-h light-dark cycles and were given access to bacteria-free food and water ad libitum. Animal handling and experimental procedures were conducted in accordance with protocols approved by the Ethics Review Committee for Animal Experimentation of Mie University (#27-6-2).

### 4.2. Induction of Polymicrobial Sepsis

A mouse sepsis model was induced by the cecal ligation and puncture (CLP) procedure as previously described [69,70]. In brief, mice were anesthetized with isoflurane and underwent midline laparotomy. The position at 50% of the entire cecal length was ligated with 6-0 sterile nylon sutures (Natsume Seisakusho, Tokyo, Japan) and an 18-gauge needle (Terumo, Tokyo, Japan) was then inserted into the edge to make a single puncture. Mice in the control group were subjected only to laparotomy, but not to CLP procedures. In both the CLP and control groups, the incision was closed by suturing and the mice were then subcutaneously infused with 1 mL of sterile saline (0.9% sodium chloride). Administration of analgesics or antibiotics was avoided due to potential influences on altering mortality or immune responses in the CLP models [71,72].

### 4.3. Isolation of Peripheral Blood Mononuclear Cells (PBMCs)

PBMCs were isolated from the blood of septic mice 24 h after CLP as previously described with some modification [73]. Briefly, heparinized blood was incubated with ammonium-chloride-potassium (ACK) lysing buffer for 5 min. The cells were pelleted by centrifugation at 300× *g* for 5 min, subjected to a resuspension in ACK buffer again, and processed as above. The cells pelleted were then washed twice with PBS by centrifugation at 300× *g*.

### 4.4. DSS-Induced Colitis

Mice were given 2.5% dextran sulfate sodium (DSS) (*w*/*v*, MW = 36,000–50,000; MP Biomedicals, Solon, OH, USA) in drinking water for seven days. Control mice were given DSS-free water. Inflammatory symptoms including body weight, bloody stool, rectal bleeding, and diarrhea were monitored daily.

### 4.5. EV Isolation via Differential Ultracentrifugation (UC)

EVs can be classified into three major subtypes—exosomes (30–150 nm), microvesicles (MVs) (50–1000 nm), and apoptotic bodies (ABs) (1–5 µm)—based on their size and biogenesis [74]. In this study, we isolated sEVs (exosomes) by eliminating MVs and ABs as described previously [75]. To avoid any confusion in nomenclature [76], we use the term “EVs” instead of exosomes or small EVs (sEVs); hereafter, the word “EVs” is used throughout this manuscript. EVs were isolated from gut-lavage fluids of control and CLP mice using differential UC as described previously with minor modifications [77]. In brief, mice were deprived of food 24 h prior to euthanasia and organ harvest. The luminal contents of the large and small intestines were obtained separately by gently flushing the lumen with phosphate-buffered saline (PBS). The lavage fluids were collected, pooled and subjected to sequential low- and high-speed centrifugation as follows. Three rounds of centrifugation at 1000× *g* at 4 °C for 10 min were done to remove insoluble materials and debris. The supernatant was centrifuged three times at 5000× *g* at 4 °C for 20 min to remove apoptotic bodies. To remove microvesicles, supernatant was transferred to ultracentrifuge tubes (Beckman Coulter, Brea, CA, USA) and centrifuged at 10,000× *g* at 4 °C for 40 min using an L-60 ultracentrifuge (Beckman Coulter). Subsequently, the supernatant was transferred to new ultracentrifuge tubes and ultracentrifuged at 100,000× *g* at 4 °C for 120 min. To purify the EV fraction, the pellet was resuspended in sterile PBS, filtered through a 0.22-μm Millex-GP syringe filter unit (Merck, Darmstadt, Germany) and ultracentrifuged again as described above. The EV pellet was resuspended in PBS and protein concentration was measured using the bicinchoninic acid (BCA) assay kit (Thermo Fisher Scientific, Waltham, MA, USA) with an iMark^TM^ Microplate Reader (Bio-Rad, Hercules, CA, USA) at 570 nm wavelength. EV samples were aliquoted and stored at −80 °C until use.

### 4.6. EV Isolation via Cushioned-Density Gradient Ultracentrifugation (C-DGUC)

In some experiments, C-DGUC was performed as described previously with some modifications [34,55]. In brief, gut-lavage fluids were pooled from sham or CLP mice (10 mice in each group) and concentrated into a final volume of 30 mL using the Amicon ultra-15 centrifugal filter unit (Merck) per the manufacturer’s instructions. The concentrated samples were placed on 0.8 M and 2 M sucrose cushions and centrifuged at 100,000× *g* for 120 min at 4 °C in as SW32Ti rotor using the L-80 ultracentrifuge (Beckman Coulter). The interface layer was harvested, diluted five-fold with HEPES buffered saline (HBS), filtered through the 0.22-μm filter unit (Merck) and then repeated for sucrose ultracentrifugation in a SW41Ti rotor as described above. The interface layer was harvested and diluted in 2.3 mL of HBS and mixed with an equal volume of 60% iodixanol to achieve a 30% iodixanol concentration containing EVs. This solution was overlaid with 20% and 5% iodixanol solutions and centrifuged at 200,000× *g* for 120 min at 4 °C in an SW32Ti rotor. Ten consecutive 1-mL fractions were diluted in PBS and centrifuged at 100,000× *g* for 120 min to pellet the EVs. Each pellet was resuspended in 100 µL of PBS and the protein concentration was measured as described above.

### 4.7. EV Measurements for Particle Size and Number

The particle size and concentration of isolated EVs were assessed using a NanoSight LM10 microscope and nanoparticle tracking analysis (NTA) software (Malvern panalytical, Malvern, UK) or by a dynamic light scattering (DLS) device (Horiba, Kyoto, Japan).

### 4.8. Flow Cytometry Analysis of EVs Conjugated to Microbeads

EVs were immobilized onto poly-L-lysine latex beads (4 µm) (Thermo Fisher Scientific) and stained with fluorophore-conjugated monoclonal antibodies as previously described [78]. Antibodies to CD63 (NVG-2), CD41 (MWReg30), CD45 (30-F11), rat IgG2b isotype (RTK4530), and mouse IgG1 isotype (MOPC-21) were purchased from BioLegend (San Diego, CA, USA). Antibodies to CD9 (KMC8) and rat IgG2a isotype (R35-95) were obtained from BD Biosciences (San Jose, CA, USA). Antibody to EpCAM (G8.8) was purchased from eBioscience (San Diego, CA, USA). The stained EVs were analyzed using a BD Accuri^TM^ C6 Flow Cytometer and BD Accuri C6 software (BD Biosciences).

### 4.9. Endotoxin Assay

EVs were tested for lipopolysaccharide (LPS) contamination using an ELISA kit (CUSABIO, Wuhan, China) according to the manufacturer’s instructions. The LPS level of EVs at an equal protein concentration (0.5 mg/mL) was measured as previously described [55]. The optical densities and LPS concentrations were analyzed at dual wavelength absorbance (detection at 450 nm and correction at 570 nm) using an iMark^TM^ Microplate Reader (Bio-Rad).

### 4.10. Mouse Ileal-Loop Assay

The biological activity of CLP and control EVs on intestinal cells was assessed in vivo using a slightly modified protocol of the ileal loop assay in accordance with a previous report with slight modifications [79]. Briefly, a midline incision (laparotomy) was performed on 2.5% DSS-treated (for seven days) mice. An approximately 4 cm-long intestinal loop was made by double ligation of the ileum using 6-0 sterile nylon sutures (Natsume Seisakusho). The 50 µg of EVs were injected at a volume of 200 µL per loop using a 29-gauge needle (Terumo) [80]. In our preliminary study, mortality was recorded 20 h following ligation of ileal tissue in mice (data not shown). Therefore, for subsequent EV treatment into ligated ileal loops, mice were euthanized 18 h post-EV treatment as described previously [81] and ileal-loop tissues were harvested for further analysis.

### 4.11. Isolation, Culture, and EV Treatment of Mouse Primary Intestinal Epithelial Cells

Primary epithelial cells were isolated as described previously [82] with slight modifications. Briefly, small intestines were harvested from 4 mice, opened longitudinally, and washed extensively with RPMI1640 medium (Nacalai, Kyoto, Japan) after mesentery, fats, Peyer’s patches, and luminal content were removed. The intestines were cut into pieces and shaken gently in RPMI-1640 containing EDTA (2 mM) and 10% fetal bovine serum (FBS) (Equitech-Bio, Kerrville, TX, USA). The tissue preparations were filtered with 70-μm mesh filters. Using 25%, 40%, and 75% Percoll (GE Healthcare Life Sciences, Chicago, IL, USA), the whole cells were spun in a centrifuge (AX-511) (Tomy, Tokyo, Japan) at 780× *g* for 20 min and IECs were obtained from the interface between the 25% and 40% layers. After verification of their expression of epithelial marker, the IECs were seeded in 12-well culture plates (Corning, Glendale, AZ, USA) at 4 × 10^5^ cells/mL in RPMI1640 containing penicillin/streptomycin and EV-depleted FBS and incubated at 37 °C with 5% CO_2_ for 12 h. Then, the cells were treated with the indicated concentrations of EVs for 24 h after which RNA was extracted for further analysis.

### 4.12. EV miRNA Analysis via Deep Sequencing

Total RNA was extracted from EVs with TRIzol reagent (Thermo Fisher Scientific). Library construction of small RNAs (including miRNAs) was carried out using an Ion Total RNA-Seq Kit v2 (Thermo Fisher Scientific) according to the manufacturer’s instructions at the Mie University Center for Molecular Biology and Genetics (Tsu, Japan). Sequencing of small RNA libraries was done using the Ion PGM system (Thermo Fisher Scientific) and data were collected using Torrent Suite v4.0.1 software. miRNA expression was shown as RPKM and miRNAs with a value of ≥1 RPKM were chosen to further assess their expressions [83]. Fold changes were calculated by taking the ratio of the individual miRNA expression values. Those miRNAs with value >1 of logarithm2 were classified as upregulated and those with <−1 as downregulated. In this measurements, the fold changes were shown with respect to the CLP group; thus, the miRNAs were upregulated or downregulated compared to the control group.

### 4.13. Gene Ontology (GO) Analysis

Target genes for upregulated miRNAs were selected based on annotations provided in the miRTarBase database (http://mirtarbase.mbc.nctu.edu.tw/php/index.php). Due to the large number of predicted targets, only those already validated by a wet lab experiment (with support type being functional) were retained and used for further study. Targets predicted from miRTarBase for the upregulated miRNAs and their targets were used to perform GO enrichment analysis using the online tool DAVID (https://david.ncifcrf.gov/). For each enrichment analysis, a set of all the known miRNA targets (as compiled in miRTarBase) was used as background in order to eliminate biases caused by non-specific enrichment of pathways resulting from miRNAs in general. The GO terms with a *p*-value of less than 0.05 were considered significant.

### 4.14. RT-qPCR

RNA (approximately 1 µg per reaction) extracted from PBMCs, mouse ileal tissues, cultured primary IECs, or EVs using TRIzol reagent was subjected to a Prime Script RT reagent Kit (Takara Bio, Shiga, Japan) or a Mir-X miRNA First-Strand Synthesis Kit (Takara Bio) in order to detect the expressions of target genes and miRNAs, respectively, according to the manufacturers’ instructions. To examine relative gene expression, qPCR was conducted using a PowerUp SYBR Green Master Mix PCR kit (Applied Biosystems, Foster City, CA, USA) and the StepOne Real-Time PCR System (Applied Biosystems) according to manufacturers’ instructions. For endogenous controls, β-actin and U6 were used to normalize mRNA and miRNA expressions, respectively [78]. For miRNAs, universal primer for the reaction was utilized as the reverse primer (Thermo Fisher Scientific). The PCR primers used in this study are listed in Appendix A. Relative expression was calculated using the comparative threshold (CT) method (2^−^^ΔCT^ for mRNA and 2^−^^ΔΔCT^ for miRNA) normalized to endogenous controls and expressed as the fold change between groups.

### 4.15. Statistical Analysis

Data are presented as the mean ± standard error of the mean (SEM). Results were analyzed using two-tailed Student’s—test for comparison of two groups and one-way ANOVA when three or more groups were compared. *p*-values < 0.05 were considered significant. Statistical analysis was performed with Prism 8 software (GraphPad, San Diego, CA, USA).

## Figures and Tables

**Figure 1 ijms-21-08445-f001:**
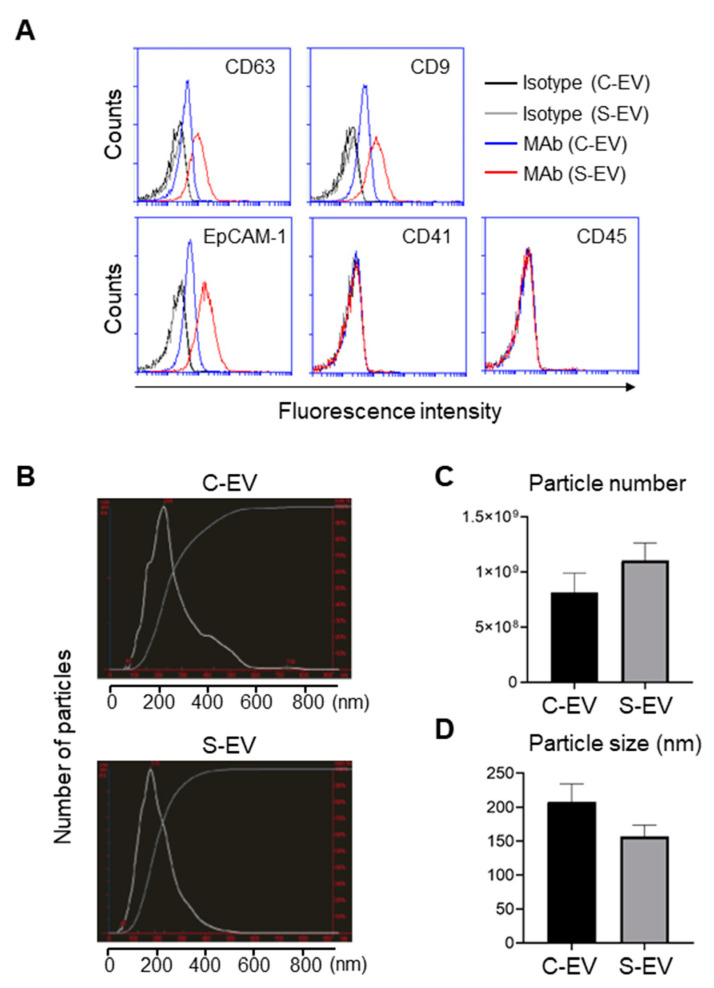
Characterization of EVs isolated from luminal washes in large intestines of control and septic mice. (**A**) Flow cytometry analysis of EVs isolated from luminal washes in large intestines of control and sepsis mice. EVs were isolated by differential UC and adsorbed on 4 μm poly-L-lysine microbeads overnight. Immobilized EVs (20 μg) were stained with indicated monoclonal antibodies (MAb) and subjected to flow cytometry to evaluate their expression. Representative histograms show changes in expression of indicated marker. Black line, isotype (C-EV); gray line, isotype (S-EV); blue line, MAb (C-EV); and red line, MAb (S-EV). (**B**) Representative NanoSight LM10 images showing sizes of EVs from control (top) and septic (down) mice. (**C**) Particle number of EVs as quantified by NTA. (**D**) Size distribution of gut- derived C-EVs (*N* = 15) and S-EVs (*N* = 16) mice groups as measured by dynamic light scattering (DLS) device. C-EV, control extracellular vesicle; and S-EV, sepsis extracellular vesicle.

**Figure 2 ijms-21-08445-f002:**
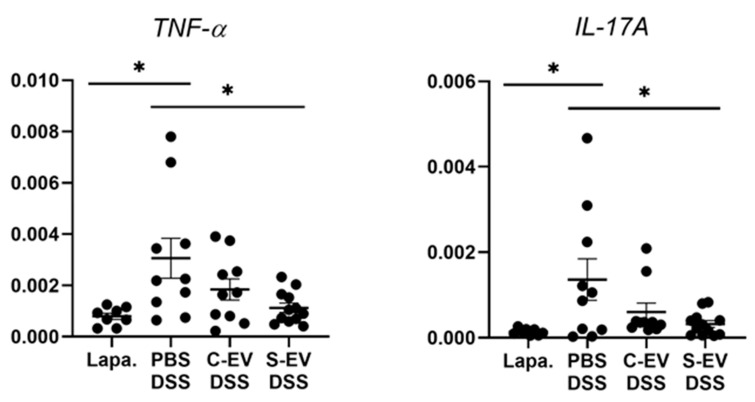
RT-qPCR analysis for gene expression of TNF-α and IL-17A. Gut inflammation was induced in the mice by 2.5% DSS-containing drinking water for seven days. Direct injection of either vehicle or differential UC-isolated EVs into the lumen of the tied ileum was done on day 7 and the tissues were separated after 18 h for RNA extraction. Relative expression of genes to *β-actin* in ileal-loop tissues of mice was accessed. Lapa, laparotomy; PBS, phosphate-buffered saline (200 μL); C-EV, control EVs (50 μg/200 μL) injected; S-EV, sepsis EVs (50 μg/200 μL) injected into ileal space; and DSS, dextran sulfate sodium. Other abbreviations: TNF-α, tumor necrosis factor-α; IL-17A, interleukin-17A. *N* = 8–12. * *p* < 0.05.

**Figure 3 ijms-21-08445-f003:**
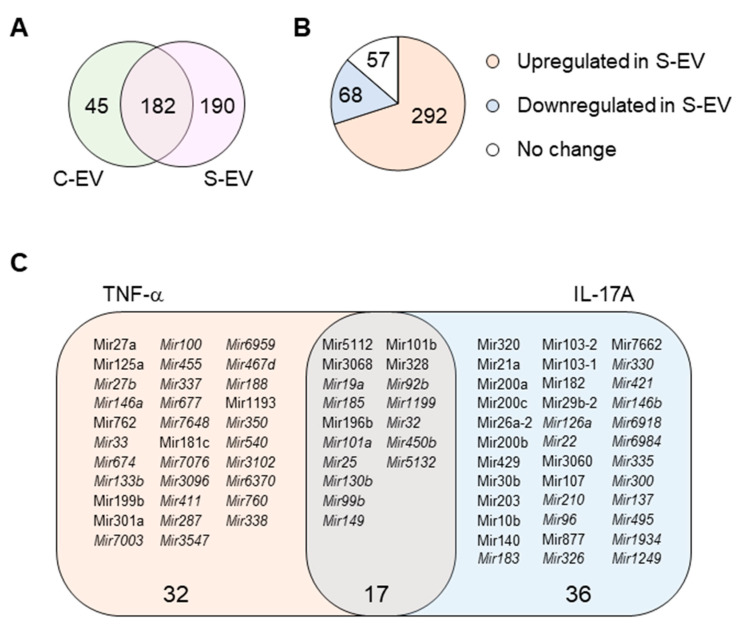
miRNA profiling of gut luminal EVs. (**A**) Venn diagram of microRNA distribution in control and septic gut luminal EVs. (**B**) Pie chart of differential expression of gut luminal EV microRNAs based on fold-change (S-EV/C-EV) in miRNA RPKM values. (**C**) TNF-α- and IL-17A-targeting miRNAs among all the sepsis-upregulated miRNAs. The miRNAs upregulated during sepsis indicate those which showed more than a two-fold increase in RPKM values in S-EV compared with C-EV. All miRNAs were chosen based on a criteria of >1 RPKM value. The miRNAs detected only in S-EVs are in italic. S-EV, sepsis EVs; and C-EV, control EVs.

**Figure 4 ijms-21-08445-f004:**
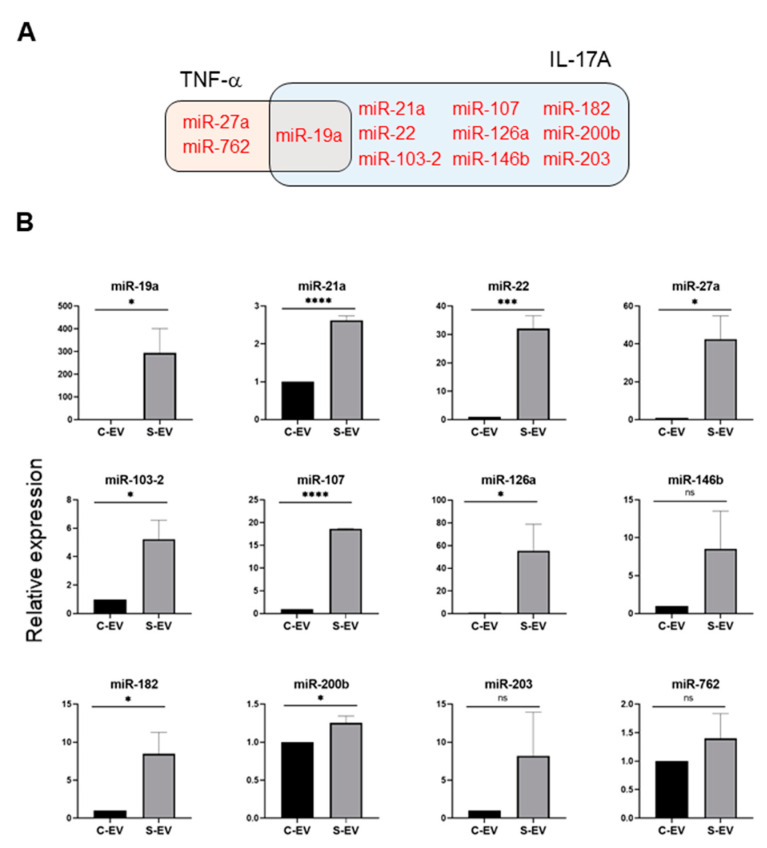
Sepsis EVs (S-EVs) enrich 12 chosen miRNAs expected to target *TNF-α* and *IL-17A*. (**A**) Among sepsis-augmented miRNAs in S-EVs, 12 miRNAs shown in Venn diagram were selected to further analyze their expressions. (**B**) RT-qPCR analysis for miRNA expressions in the EVs. These miRNAs were tested for their increased enrichment in S-EV. As a control, U6 was used to normalize miRNA levels. Total RNAs were isolated from EVs pooled from 5 mice in each group. At least four separate experiments were performed. Results are shown as the mean ± SEM. S-EV, sepsis EVs; and C-EV, control EVs. * *p* < 0.05; *** *p* < 0.001; **** *p* < 0.0001; and ns, not significant.

**Figure 5 ijms-21-08445-f005:**
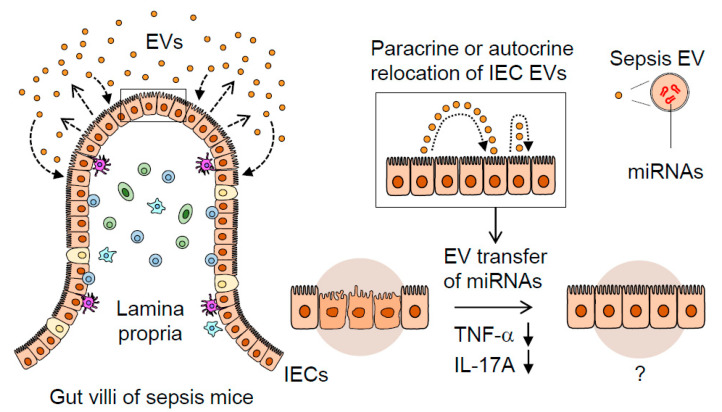
A proposed mechanism illustrating the dynamic role of septic IEC-derived luminal EVs in regulating gut inflammation. Sepsis induces gut-barrier dysfunction and epithelial injury. Sepsis-induced luminal release of EVs increases their epithelial trait. The luminal EVs possessing different miRNA expression levels relocate themselves to IECs in either a paracrine or autocrine fashion. Thus, the EVs secreted from IECs are thought to contribute to dampening pro-inflammatory responses in sepsis-damaged gut mucosa via the transfer of regulatory miRNAs.

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
