# Peer review of "Intestinal Epithelium-Derived Luminally Released Extracellular Vesicles in Sepsis Exhibit the Ability to Suppress TNF-α and IL-17A Expression in Mucosal Inflammation"

_ijms, 2020, doi:10.3390/ijms21228445_

Round 1
Reviewer 1 Report
In this original article authors have isolated the small extracellular vesicles (exosome) from sepsis mice and investigated the ability of these EVs to suppress the inflammation in colitis mice. The characterisation of EVs were done by well-established technique (flow cytometry and nano particle tracking analysis). A lot of literature suggests that EVs isolated from mesenchymal stromal cells exert beneficial activity for a range of disease while EVs isolated from diseased condition/cells (example cancer cell, inflammatory condition) cause damage which primarily depends on the cargo (mRNA, miRNA, DNA) they transfer from donor to recipients’ cells. In contrast, this study highlights that the septic EVs treatment to inflamed gut showed significant reduction in pro-inflammatory cytokines. The topic is very interesting and falls under the scope of the journal. However, authors need to revise the manuscript substantially and address the comments below before considered appropriate for publications.
Comments
- Section 4.9 “Mouse Ileal-loop assay: Authors have mentioned at line 408-409 “The EVs were injected at a volume of 200 µL per loop using a 29-gauge needle (Terumo)”
The dose of EVs is not clear. If authors have isolated EVs and quantified for protein concentration by BCA assay, then it will be more relevant to mention the exact dose of EVs rather than just saying 200 µL.
How did authors decided to give this 200 µL of EVs? Is this clinically relevant dose, please cite appropriate paper (if any).
- What was the purpose of isolating EVs by two different methods (section 4.4 Ultracentrifuge and section 4.5 C-DCUC).
- The induction of mucosal inflammation is very confusing. Somewhere author have mentioned mice were given with 2.5% of DSS for 6 days line 146) in water, and somewhere it is mentioned 7 days (line 166, 344, 407). Please clarify this.
- Authors treated EVs for 18 hours and sacrificed the mice. Any reason for just 18 hours treatment? Please cite appropriate paper for this or provide appropriate rationale.
- Results: figure 2, is this result from UC isolated EVs or C-DCUC isolated EVs?
- Any relevant information on “if the EVs released during sepsis alters the profile of gut microbiome” should be discussed.
- I would recommend authors to add more results to correlate in vivo with in vitro. Example: It would be interesting to see the effect of these sepsis derived EVs on mouse primary macrophage or other relevant primary cells to see the level of TNF-alpha and IL17A or other inflammation mediators.
- Text are cryptic in many places. Needs correction for structure, spelling, meaningful/clarity of sentences.
Author Response
Reviewer 1
In this original article authors have isolated the small extracellular vesicles (exosome) from sepsis mice and investigated the ability of these EVs to suppress the inflammation in colitis mice. The characterisation of EVs were done by well-established technique (flow cytometry and nano particle tracking analysis). A lot of literature suggests that EVs isolated from mesenchymal stromal cells exert beneficial activity for a range of disease while EVs isolated from diseased condition/cells (example cancer cell, inflammatory condition) cause damage which primarily depends on the cargo (mRNA, miRNA, DNA) they transfer from donor to recipients’ cells. In contrast, this study highlights that the septic EVs treatment to inflamed gut showed significant reduction in pro-inflammatory cytokines. The topic is very interesting and falls under the scope of the journal. However, authors need to revise the manuscript substantially and address the comments below before considered appropriate for publications.
We are grateful to Reviewer 1 for the positive and constructive comments on our manuscript. We agree that the original manuscript requires further elaboration to better convey the main points. We have therefore tried to improve it by substantially revising the manuscript and provide below point-by-point responses to all of the concerns raised.
Comments
- Section 4.9 “Mouse Ileal-loop assay: Authors have mentioned at line 408-409 “The EVs were injected at a volume of 200 µL per loop using a 29-gauge needle (Terumo).” The dose of EVs is not clear. If authors have isolated EVs and quantified for protein concentration by BCA assay, then it will be more relevant to mention the exact dose of EVs rather than just saying 200 µL. How did authors decided to give this 200 µL of EVs? Is this clinically relevant dose, please cite appropriate paper (if any).
We are grateful to Reviewer 1 for raising these critical points. We have revised the manuscript by adding information on the precise dose (50 μg) of EVs used (line 443). In addition, we have carefully measured the length of ligated loops using a ruler and found it was around 4 cm. We have thus corrected from 3 to 4 cm in the revised manuscript (line 442). Regarding the scientific basis to use the volume (200 μl) for loop injection, we have decided it according to the method of a previous report (Fukuda et al., J Vis Exp 2011 58). In this paper the authors used 50-μl volume into 1-cm of ligated loop. Thus, we decided to use 4-times larger volume (200 μl) to inject 4-cm ligated loop in current study. We cited this paper as a reference in the revised manuscript (line 444).
- What was the purpose of isolating EVs by two different methods (section 4.4 Ultracentrifuge and section 4.5 C-DCUC).
We are thankful to the Reviewer for pointing out this important issue and have incorporated this information in the revised manuscript (lines 161-166). As the Reviewer indicated, we have described the purpose of isolating EVs by two different methods (lines 161-164). We have also added the explanation in the revised manuscript (lines 164-166) and put two relevant papers as references reporting that this C-DGUC method is considered offering credible EV purity. In addition, we corrected the word “conventional UC” to “differential UC” (lines 88, 121, 164, 179, 384, and 390) because the latter has been considered as the scientific terminology used in previous reports (Helwa et al., 2017 PLoS One PMID 28114422; Gupta et al., 2018 Stem Cell Res Ther PMID 29973270).
- The induction of mucosal inflammation is very confusing. Somewhere author have mentioned mice were given with 2.5% of DSS for 6 days line 146) in water, and somewhere it is mentioned 7 days (line 166, 344, 407). Please clarify this.
Thank you for this indication for clarifying the exact time for DSS treatment. Mice were given 2.5% DSS for 7 days. It has been corrected in the revised manuscript (line 148).
- Authors treated EVs for 18 hours and sacrificed the mice. Any reason for just 18 hours treatment? Please cite appropriate paper for this or provide appropriate rationale.
Thank you for these insightful comments. The ileal loop assay is a well-established method that has been used extensively to study interactions between the intestinal mucosa and bacteria or their toxins as well as mucosal inflammation. However, creating ileal loops leads to intestinal obstruction and death. In our preliminary experiments we observed substantial mortality in mice 20 hours after making ileal loops. Thus, we decided to euthanize mice and collect ileal tissues at 18 hours post-treatment. Moreover, studies that used the ligated ileal loop method in mice have done within such period typically not exceeding 18 hours. This relevant description for this time point has been added with a relevant paper as a reference in the revised manuscript (lines 444-447).
- Results: figure 2, is this result from UC isolated EVs or C-DCUC isolated EVs?
Thank you for this indication to clarify the precise method for EV isolation used in Figure 2. The results shown in Figure 2 are those obtained from differential UC-isolated EVs. The words “differential UC-isolated” have been added in the Figure legend (line 179) in the revised manuscript.
- Any relevant information on “if the EVs released during sepsis alters the profile of gut microbiome” should be discussed.
We thank the Reviewer for this critical suggestion and have included a description in the Discussion of the revised manuscript (lines 269-279).
- I would recommend authors to add more results to correlate in vivo with in vitro. Example: It would be interesting to see the effect of these sepsis derived EVs on mouse primary macrophage or other relevant primary cells to see the level of TNF-alpha and IL17A or other inflammation mediators.
We thank the Reviewer for these valuable and helpful comments. In accordance with these suggestions, we have performed in vitro experiments. We first isolated primary gut epithelial cells (ECs), confirmed their expression of epithelial marker EpCAM (Supplemental Figure 9A), and then treated them with different EV doses (1, 5, or 10 μg/ml) to determine the impacts on regulating TNF-α and IL-17A expression (please also see the section 4.10 for newly added methods of isolation of primary gut ECs). However, in this experimental setting, S-EVs as well as C-EVs didn’t exhibit any significant reduction in TNF-α and IL-17A expression in the primary gut ECs, when compared with vehicle treatment (Supplemental Figure 9B). Accordingly, establishment of more physiologically relevant models might be beneficial to achieve in vitro outcome that correlates with in vivo result. With regard to these finding, we have added relevant descriptions and speculations in the revised manuscript (lines 167-175 and 330-340).
- Text are cryptic in many places. Needs correction for structure, spelling, meaningful/clarity of sentences.
We thank the Reviewer for this important suggestion. We have accordingly corrected many cryptic words, phrases, and sentences in the revised manuscript. Thank you, again.
Reviewer 2 Report
- Studies have shown that analgesics are very important to minimize the pain and make the mice to survive better. I would like to know from the author, how they managed to keep the CLP mice alive without administration of analgesia or antibiotics.
- How the author confirmed the induction of CLP in those mice?
- For figure1, n number is low, but the standard error is also very less. How come the author can get such a low standard error using such a low n number. I would rather suggest the author to add more mice in this expt.
- Figure 4 also has very low n number.
- Though the author gave references to show that female mice showed variability in sepsis, but my concern is about the further use. If I agree with the author that this extracellular vesicles in sepsis exhibit the ability to suppress TNF-alpha and IL-17A expression in mucosal inflammation, what about the next step for their study. Doesn’t the author want this protocol to work in human in near or far future? In that case, what will happen with the female patients if they are not able to show any data on female mice/higher animal/human subject at all. There are also many articles showing CLP consistent data in both male and female mice. Authors are suggested to consider those articles too and add both male and female in the expt.
Author Response
Reviewer 2
- Studies have shown that analgesics are very important to minimize the pain and make the mice to survive better. I would like to know from the author, how they managed to keep the CLP mice alive without administration of analgesia or antibiotics.
We thank the Reviewer for pointing out these important issues. We agree with the relevance of analgesia or antibiotic administration in the mouse CLP sepsis model. However, as described previously, we realized that analgesia and antibiotics may also have confounding effects on immune response in the septic mice (lines 370-373)(Refs. 71 and 72). With respect to how the mice survived, we supplied physiological saline solution (lines 373-375) after surgery to mice, and as shown in Supplemental Figure 1 the mice had high survival within the 24-hour time frame of our study. In most of our subsequent studies, the mice subjected to this CLP model had 100% survival 24 hours after surgery.
- How the author confirmed the induction of CLP in those mice?
We thank the Reviewer for raising these critical points and have included a description in the revised manuscript accordingly (lines 90-95). Additionally, we have isolated mouse peripheral blood mononuclear cells from sham (control) and septic mice and analyzed a sepsis-increased expression in messages of some pro-inflammatory cytokines. We have included the new method (lines 374-379), the results (Supplemental Figure 2), and the description (lines 92-95) in the revised manuscript.
- For figure1, n number is low, but the standard error is also very less. How come the author can get such a low standard error using such a low n number. I would rather suggest the author to add more mice in this expt.
We are grateful to the Reviewer for raising these critical points. In accordance with these suggestions, we have conducted the DLS analysis to increase n number using EV samples newly and individually isolated from several mice and revised the data in the Figure 1. In addition, both graphs to show size and number of particles were assessed again using GraphPad Prism8 software, although the particle numbers were shown as the original data due to the current unavailability to NTA. However, we increased mouse number (control, N=15; and sepsis, N=16) to obtain more convincing data of EV size in this revised manuscript.
- Figure 4 also has very low n number.
We thank the Reviewer for this important indication and we have accordingly performed new Q-PCR analysis with additionally isolated EV samples. We have changed the Figure 4 in the revised manuscript.
- Though the author gave references to show that female mice showed variability in sepsis, but my concern is about the further use. If I agree with the author that this extracellular vesicles in sepsis exhibit the ability to suppress TNF-alpha and IL-17A expression in mucosal inflammation, what about the next step for their study. Doesn’t the author want this protocol to work in human in near or far future? In that case, what will happen with the female patients if they are not able to show any data on female mice/higher animal/human subject at all. There are also many articles showing CLP consistent data in both male and female mice. Authors are suggested to consider those articles too and add both male and female in the expt.
We are thankful to the Reviewer for the insightful comments. We agree with the Reviewer’s profound view, especially in the aspects of further use, which should be considered rightly regardless of gender. We have corrected the relevant description accordingly in the revised manuscript (line 358). In addition, we would like to mention that the EVs isolated from intestinal lavage fluids of both male and female mice were used in EV characterizations as well as miRNA analysis.
Round 2
Reviewer 1 Report
Thank you for addressing all of my comments. The quality of revised version of the manuscript looks good. I have recommended for publications.
Reviewer 2 Report
I would like to thabk the author for addessing the raised concern and for making the article more interesting after the addition. No further comments.